# Methods of Betulin Extraction from Birch Bark

**DOI:** 10.3390/molecules27113621

**Published:** 2022-06-05

**Authors:** Olga V. Demets, Altynaray T. Takibayeva, Rymchan Z. Kassenov, Madina R. Aliyeva

**Affiliations:** 1Department of Chemistry and Chemical Technologies, NJSC Karaganda Technical University named after Abylkas Saginov, Karaganda 100027, Kazakhstan; altynarai81@mail.ru (A.T.T.); madiko8707@mail.ru (M.R.A.); 2Department of Organic Chemistry and Polymers, Chemistry Faculty, NJSC Karaganda University named after Y.A. Buketov, Karaganda 100024, Kazakhstan; r_z_kasenov@mail.ru

**Keywords:** betulin, birch bark, pentacyclic triterpenoid, biological activity, methods of betulin isolation

## Abstract

Betulin is the most popular of the known triterpenoids of the lupan series. It has valuable pharmacological properties. It exhibits antibacterial, antiviral, antitumor, hypolipidemic and other types of activity. The prospects of using betulin in medicine, pharmacology, and veterinary medicine require the development of effective methods for obtaining it from waste from the woodworking industry. Therefore, the question arises of the need to develop a technology for isolating and purifying triterpenoids from birch bark in large quantities. This review contains a variety of methods for the isolation of betulin. The advantages and disadvantages of the proposed methods are described. The following methods are considered: vacuum sublimation after preliminary alkaline; vacuum pyrolysis; supercritical extraction with carbon dioxide or mixtures of carbon dioxide with various solvents; extraction with organic solvents. Also, the method of microwave is described, it is activation on the example of the Kyrgyz birch (*Betula kirghisorum*), this is an endemic species that has not been studied before, growing on the territory of the Republic of Kazakhstan.

## 1. Introduction

Betulin (1) is a natural pentacyclic triterpenoid belonging to the lupan group and having the systematic name 3β,28–dihydroxy-20(29)-lupen or lup-20(29)-en-3β,28-diol (Figure 1). Betulin (betulinol, birch camphor, lupendiol) is a crystalline organic substance discovered in 1778 by Toviy (Johann Tobias) Lovitz, who, by sublimation, isolated a white compound from birch bark and for the first time described its chemical properties. Later, in 1831, a famous chemist of that time, D. Mason, gave it the name betulin (from the Latin word ‘betula’), which is also contained in birch tar and is a white resinous substance that fills the cavities of cork tissue cells on birch trunks and gives it a white color [1].

The structure of the lupine-series triterpenoids is based on a complex polycyclic system with a cyclopentane perhydrophenanthrene (sterane) core consisting of four cyclohexane and one saturated cyclopentane rings condensed together. In accordance with the accepted numbering of the carbon skeleton atoms and the increasing numbers of carbon atoms, the rings have the designations: **A**, **B**, **C**, **D** and **E** (Figure 1). A characteristic sign of belonging to this group of betulin is the presence of a five-membered ring **E** and an *α*-isopropenyl group in the C-19 carbon atom [2,3,4].

It has valuable pharmacological properties. It exhibits antibacterial, antiviral, antitumor, hypolipidemic and other types of activity. Up-to-date information on the use of triterpenes in medicine and in the national economy is given in [5], where much attention is paid to betulin and its derivatives, and their relatively easy availability is emphasized. Thus, the availability and wide range of biological activity of betulin place it as one of a number of valuable natural sources for use both in its native state and in the form of various transformation products. The prospects for the versatile use of betulin in medicine, pharmacy, veterinary medicine, and the paint and varnish industry require the development of effective methods for its production from the waste-wood processing industry. This review combines many existing methods of isolation and systematizes them into a single scientific publication. The review describes the most promising methods of betulin extraction from birch bark.

## 2. The Prevalence of Betulin in Nature

Betulin is found in warty birch, or hanging birch *(Betula verrucosa Ehrh*. *or Betula pendula Roth.)*, and fluffy birch (*Betula rubescens Ehrh*.) widely distributed in the Northern Hemisphere [6]. Betulin (Figure 1) is also found in the birch bark of white birches *(Betula alba)* growing in Europe [7]. Although betulin has been found in at least two dozen plants belonging to various genus and families, the main source of betulin is birch bark. Fagaceae, Rosaceae, Rhamnaceae, Dilleniaceae, Platanaceae, Betulaceae are the plant families from which betulin and betulinic acid were isolated [8]. For example, betulin, in addition to birch bark, is found in hazel bark, calendula and other medicinal plants. Examples have been given when betulin, lupeol and betulinic acid have been isolated from *Ziziphus jujuba* [9,10], as well as from the ground part of thistle (*Atractylis carduus)* [11]. Betulinic acid is found in the leaves of plumeria *(Plumeria obtusa)* [12], orchid *(Orchid Lusiaindivisa)* [13] and other plants *(Dillenia papuana, Tryphyllum peltatum, Ancistrocladus heuneaus, Diospyrus leucomelas, Tetracera boliviana, Sizyphus joazeuro, Syzigium claviflorum, Aerua javanica)* [14,15,16,17,18].

Birch bark has two clearly distinguishable parts, external and internal. The outer part of the bark is the most rich in extractive substances. The main component of almost all extracts is betulin, which determines the white color of the bark. The variety of properties of betulin is determined by the philosophy of the origin of this substance: plants synthesize betulin to protect against all adverse environmental factors and accumulate it exclusively in their shell. In the composition of plant membranes, betulin is necessary to protect the plant from damaging environmental factors: radiation, bacteria, fungi, viruses, and insects [19].

## 3. Methods of Betulin Isolation

Birch bark is the best raw material for the isolation and production of betulin, not only in terms of its content, but also because of the large amount of bark formed by reverse industrial waste, especially the pulp and paper industry. Currently, birch bark is mainly burned to produce heat and electricity instead of being used more efficiently as a source of triterpenes [20,21,22].

The amount of betulin can be up to 20–30% (or even almost 45%) [23] of the dry mass of the outer bark, depending on the species of birch (biological species) and its regional location [24,25,26]. It has been claimed that much less betulin and related compounds are contained in the root and leaves of birch [27]. Betulone, erythrodiol, and oleanolic acid may be contained in significantly smaller amounts in birch bark [14,28,29]. The common European species *Betula pendula Roth.* and *Betulaceae (Betula alba)* may contain 10–20% lupeol and betulinic acid relative to the content of betulin in the outer bark, and both of these triterpenes are the main impurities of raw betulin [30]. 

Due to its pronounced antifungal and antimicrobial properties [31,32,33], betulin has the ability to protect birch from fungal and bacterial damage through its bark.

It is well-known that triterpenes can be extracted with good yields (up to 40%) from birch bark by various organic solvents [34,35]. Although the calorific value of the outer bark of birch exceeds 30 MJ/kg [36], processing it into commercial products with a higher added value seems to be a more promising way than using it as fuel. This circumstance determines the need to develop a technology for isolating and purifying triterpenes from birch bark in large quantities, using simple and environmentally friendly methods, the arsenal of which is currently insufficiently representative.

The steadily growing interest in betulin and its derivatives is primarily due to the very wide range of its applications in various fields [37,38]. The availability and biological activity of betulin make it one of a number of valuable natural compounds; therefore, it is currently an urgent task to develop convenient and progressive methods for extracting betulin.

The effectiveness of the extraction of betulin and its derivatives makes it possible to rationally use birch as a plant raw material, which will allow the finding of new and unexpected areas of application of such a fertile renewable source of natural raw materials for human life. 

In recent decades, many works have appeared on the methods of obtaining and purifying betulin. Most of the known methods of isolation (Figure 2) [39,40,41,42,43,44,45,46] of betulin are based on methods of birch-bark extraction with various solvents. At the same time, the resulting extract contains, as a rule, in addition to betulin, a significant range of other compounds, such as lupeol, betulin aldehyde, and lupeol aldehyde, and needs additional purification from them [19,47].

Known methods of the isolation of betulin from birch bark involve recrystallization of its extract, and the applied purification methods are divided into two groups [47]. In the first group, birch-bark extraction is carried out with various organic solvents—chloroform in the presence of silica gel, 90–95% ethyl alcohol in the presence of activated carbon [48], toluene [49], or acetone [50]. The second group includes methods involving the use of alkali to separate the acidic components of the extract: (a)Birch bark is treated with water–alcohol alkali (alcohol: MeOH, EtOH, *i*-PrOH);(b)An extract is washed with an alkali solution of different concentrations [51,52,53,54]

It is claimed that the maximum yield of betulin is achieved only with the alkaline hydrolysis of birch bark, which proceeds under sufficiently harsh environmental conditions. In general, the technological process of betulin extraction is shown in Figure 3.

The crushed birch bark is loaded into the extractor (1), where a solvent is fed from the collector (2) by a pump (3), which is heated in the heater (4). Solvent vapors condense in the refrigerator (5), and the duration of extraction is 5–6 h. After extraction, the hot extract is poured into the crystallizer (6), the solvent residues are driven away from the bark with sharp steam, the solvent is separated in florentine (7) and enters the collector (2). The crystallizer is equipped with a stirrer and a cooling jacket in various designs. After cooling, the extract is fed to a vacuum nutch filter (8), and the filtrate goes through an intermediate receiver (9) for evaporation into the evaporation apparatus (10). At the final stage, the solvent is removed from the precipitate by sharp steam [55].

When using a mixture containing petroleum ether (fraction 70–100 °C) and 30–75% toluene as an extractant, the extract yield is from 16 to 25% of the absolutely dry substance, and the content of betulin in the sediment is from 90 to 95% [55].

The disadvantages of the known methods of obtaining betulin include the need for the operation of the preliminary separation of bark into birch bark and bast, which is quite time-consuming and energy-consuming. In addition, only up to 20% of the mass of the entire bark is processed, since the share of the outer layer of birch bark accounts for 16–20%. In addition, there are huge resources of birch bark in which there is no clear boundary between its outer and inner parts. Such bark is found in the trunks of overgrown trees, as well as in clumps of fairly young birch trees [56].

In order to increase the yield of extractives and reduce the duration of extraction, various methods of intensification are used. Some of them improve the hydrodynamic regimes of the extraction process [57], others are based on the use of the mechanical, chemical and mechanochemical activation of raw materials [58,59]. In some cases, a positive effect is achieved by fractional extraction of tree bark with organic solvents of different polarity. 

It is known that one of the effective methods of the extraction of plant materials is microwave processing in an ultrahigh frequency (microwave) field. The nature of the effect of the microwave field is similar to intensive moisture–heat treatment carried out by combining the effects of conductive heating and sharp steam, but the destruction of the birch-bark structure under the action of the microwave field occurs to a greater extent. The effect of the microwave field creates excessive pressure in certain areas of the plant structure due to the vaporization and expansion of air in capillaries and voids. 

Paper [60] shows the possibility of intensifying the process of betulin isolation from the birch bark of the Kirgyz birch using microwave exposure. The study of methods of betulin isolation from the birch bark of the Kirgyz birch *(Betula kirghisorum)* depend on the extraction conditions.

In order to determine the effect of the alkali concentration in the alcohol solution, the duration of hydrolysis and the increase in the yield of betulin from the birch bark of the Kirgyz birch, a series of experiments were conducted, the results of which are presented in Table 1.

As can be seen from the data presented in Table 1, the maximum yield of betulin under the conditions of the classical hydrolysis method is observed in a 20% alkali solution for 3 h. All betulin samples extracted with butanol from the birch bark of the Kyrgyz birch were hydrolyzed in the presence of sodium hydroxide and recrystallized from ethanol.

The microwave extraction method was used to intensify the process of betulin isolation from the birch bark of the Kirgyz birch. Data on the yield of betulin extracted using a microwave field in the presence of sodium hydroxide from the birch bark of the Kirgyz birch are given in Table 2.

As can be seen from Table 2, under the conditions of the classical hydrolysis method, the maximum betulin yield of 10.9% is observed when processing birch bark for 3 h. Under the conditions of exposure to a microwave field at an alkali concentration of 20–25%, the yield of raw betulin is 8.47% and 9.13% within 6 min. The highest yields of betulin (15.55–18.25%) are observed when exposed to a microwave field for 9 min. Consequently, compared with classical methods of betulin extraction, the extraction rate in the microwave field increases by 15–20 times. With a further increase in the duration of the extraction process, the yields of the target product decrease, which, apparently, is associated with the course of betulin tarring processes.

Raw betulin, obtained by the method of microwave extraction, is a light gray powder that is odorless and with foreign inclusions. After recrystallization from isopropyl alcohol, betulin acquires a characteristic white color.

Thus, in this work, the dependence of the quantitative yield of betulin from Kirgyz birch on the duration of extraction and the concentration of an aqueous alkali solution was studied. The maximum yield of betulin under the conditions of the classical hydrolysis method is observed when boiling birch bark of the Kirgyz birch for 3 h in an aqueous alcohol solution of alkali. With the intensification of the betulin-extraction process from the birch bark of the Kirgyz birch using the microwave extraction method, the best betulin yields are observed when exposed to a microwave field for 9 min. Compared with classical methods of betulin extraction, the extraction rate in the microwave field increases by 15–20 times [60].

In [61], the influence of the nature of extractants in the process of the thin-film vapor-phase extraction of birch bark on the composition and degree of extraction and the shape of the products obtained was investigated.

Extraction is one of the main ways of separating plant raw materials into individual components or their complexes. Extractants are selected based on the physico-chemical properties of the extracted compounds. By selecting the appropriate solvent, the selectivity of extraction can be achieved.

The most commonly used methods are simple in execution and are effective: maceration, digerization, percolation, perforation and extraction [48]. Very often, it is impossible to draw a clear line between these processes. The technological and hardware design of the extraction process is diverse: from the use of Soxlet’s devices in all its varieties and percolation in dividing funnels, to the most widespread extraction method: carrying out consecutive single treatments of a portion of raw materials with a fresh solvent at normal temperature (20–25 °C) (maceration) or heated (digering), with the collection and combination of the obtained extract solutions with the subsequent distillation of the solvent from the extract.

The efficiency of any type of extraction of a solid by a liquid depends primarily on its solubility and the rate of transition from one phase to another. The solubility can be changed by selecting the appropriate solvent, into which the desired substance mainly passes, and the impurities present remain in the solid phase. The rate of the transition of a substance from a solid phase to a solution is determined mainly by the rate of penetration of a liquid into the solid phase, the rate of diffusion of a substance into a liquid and the rate of removal of a substance from the interface of phases. Unlike a system of two liquid phases, equilibrium at the boundary of the solid and liquid phases occurs very slowly. It is possible to accelerate the approach to the equilibrium state by increasing the surface of the solid phase by grinding the sample or by constantly feeding a fresh solvent to the phase boundary. In addition, it is possible to accelerate the achievement of equilibrium by simple mixing (with maceration and digeration) or by using counterflow (with percolation).

Based on the analysis of the shortcomings of existing extraction methods and the physico-chemical laws of the extraction process, a new highly effective extraction method has been developed—thin-film vapor-phase extraction [62]. Birch bark crushed to particles of 3–5 mm in size and dried to a constant mass at a temperature of 105 °C was used as a feedstock in this work. The extraction process was carried out in a laboratory extractor (Figure 4). The extractor is an extraction column (5); the reverse refrigerator (6) is connected from above, and the receiver (2) is connected from below with a fitting for entering hot water vapor. The receiver is heated by a heater (1). Crushed birch bark (4) is loaded into the extraction column. An extractant (a two-component mixture of a low-boiling solvent with a boiling point below 100 °C and water) is loaded into the receiver. The receiver is heated by water vapor supplied to it or by a heater. The extractant vapors pass through a layer of crushed birch bark and condense in the reverse refrigerator (6). The condensate irrigates the layer of crushed birch bark and, passing through its layer, is enriched with extractive substances and flows into the receiver. The extractant evaporates and returns to the extraction column, and the dissolved extract precipitates as a granulate (12).

After the extraction is completed, the reverse refrigerator (6) is replaced with a direct refrigerator (9) and the boiling solvent from the receiver and crushed birch bark in the extraction column is driven off with sharp water vapor or by increasing the temperature of the receiver. The precipitate is separated by filtration and dried. The regenerated low-boiling solvent is reused.

The duration of extraction was 2–4 h. The analysis of the obtained extract samples was carried out on the HP 1100 HPLC chromatograph (Agilent, CA, USA), Microbore (MZ-Analysentechnik GmbH, Mainz, Germany) 2 × 250 mm column, mobile phase: 95% CH_3_OH and 5% H_2_O, acidified with H_3_PO_4_ (calculated 0.2 mL/L H_2_O). A diode array detector (Agilent, CA, USA) with 180–700 nm range. The identification of peaks was carried out according to the standard and by UV absorption spectra. In order to establish the influence of the nature of solvents on the intensity and selectivity of extraction, the authors investigated the dependence of the yield of betulin (total extract) and the content of betulin in it directly on the nature of the extractant (Table 3).

From the analysis of the data in Table 3, it can be concluded that the choice of a low-boiling solvent largely determines both the qualitative and quantitative composition of the resulting product. Thus, when using hexane, it is possible to obtain the purest betulin to the detriment of other extractive substances. When using ethyl acetate, the qualitative composition of the extract expands, while maintaining the completeness of betulin extraction. Therefore, the selection of the composition of extractants can be controlled by the selectivity of the extraction process. It was found [19] that, with exhaustive birch-bark extraction, it is possible to achieve a betulin yield of 9.30–9.48%; however, after two hours of extraction, about 58.5–67.8% of betulin from its total content in birch bark is extracted, and after four hours the yield reaches 81.1–83.4%. Further carrying out the process leads to an increase in the completeness of the extraction of products, but, at the same time, production costs significantly increase.

In addition, it was found that the extraction temperature regime affects the ratio of the extractant components in the extraction column and in the receiver. This largely determines the qualitative composition of the resulting product and its mechanical properties.

It is important that it is the presence of water, as one of the extractant components, that contributes to the granulation of the resulting sediment in the receiver, which ultimately leads to the simplification of subsequent technological stages of processing, such as the filtration and drying of the final product. A range of ratios of water and a low-boiling solvent in the receiver should remain stable. Currently, the available information on the purification of betulin from impurities is unsystematic and mainly represents methods based on the recrystallization of betulin from polar proton and aprotic solvents. It is known [63] that the use of solvents such as lower aliphatic alcohols, chloroform, acetone, ethyl acetate, and dichloromethane for recrystallization leads to the formation of solvate complexes with betulin, which makes it difficult to identify it as an individual compound. In addition, it is necessary to note the low selectivity of these solvents relative to the impurities accompanying betulin, such as lupeol, polyphenols and oxidized forms of betulin, which are also soluble in these solvents and worsen the quality of the resulting betulin. For example, when using ethanol as a solvent for recrystallization, there are significant losses of betulin (up to 40%), which remain dissolved in the mother liquor, which is associated with a narrow range of changes in the solubility of betulin depending on temperature. Thus, for 86% ethanol, the content of betulin at boiling point is 5.35%, and at 8 °C—1.86% (i.e., the loss of betulin in the mother liquor in this case reaches 34.8%) [64].

In this regard, the aim of one of the works [65] was to search for a relatively simple method that ensures the production of betulin of a high degree of purity, suitable for further use in organic synthesis and for biological research without additional purification and minimizing betulin losses.

The total extract containing betulin was obtained by the thin-film vapor-phase extraction of birch bark *(Betula pendula Roth.)* with ethanol according to the previously proposed method [62,66].

For recrystallization, the following were used: *n*-octane, *n*-nonane, *n*-decane, *n*-undecane, commercially available “chemically pure” qualifications, without additional purification. Nefras C4–150/200 and kerosene TS–1 were pre-distilled to separate non-volatile heavy impurities and fractions with boiling points in the range of 150–200 °C were collected.

Recrystallization was performed twice. A total dry alcohol extract of birch bark containing betulin and a selected solvent was placed in a glass at the rate of 25 mL of solvent per 1 g of dry extract, the suspension was heated to a boil and boiled with stirring until the solution became transparent, while insoluble impurities remained at the bottom of the glass. Then, the contents of the glass were subjected to hot filtration, insoluble impurities were separated, washed with hexane on the filter and dried in vacuum at a temperature of 50 °C. The filtrate was cooled to a temperature of 0 °C, while a precipitate of betulin fell out, which was filtered, washed with hexane on the filter and dried in vacuum at a temperature of 50 °C. The dried betulin and the fraction of insoluble impurities were weighed. The solvent was distilled dry in the vacuum of a water jet pump and the dry cubic residue was weighed to account for the loss of betulin remaining in the mother liquor.

Dried and recrystallized betulin was subjected to repeated recrystallization by the same method. Product losses were determined in terms of pure betulin. 

The analysis of the obtained products was carried out by the HPLC method on a micro-column liquid chromatograph Milichrome A-02 (Closed Joint Stock Company Institute of Chromatography “EkoNova”, Novosibirsk, Russia) with UV detector and a 2 × 75 mm column filled with the sorbent ProntoSIL—120–5-C18 AQ (Limited Liability Company Institute of Chromatography “EkoNova”, Novosibirsk, Russia). Elution was carried out with a mobile acetonitrile/water phase in a ratio of 80:20, in an isocratic mode at a column temperature of 35 °C and an eluent flow rate of 100 µL/min. Detection was carried out at a wavelength of 210 nm in accordance with the maximum absorption of betulin. For the preparation of eluents were used acetonitrile (grade 1) from Cyochrome (St. Petersburg, Russia) and distilled water purified using the Milli-Q system (Millipore, MA, USA). The studied betulin samples were dissolved in an acetone–acetonitrile mixture in a ratio of 40:60. The volume of the aliquot was 5 µL.

The quantitative content of betulin was determined by absolute calibration. The calibration graph is described by a linear function C = 0.0924·S, where S is the area of the chromatographic peak and C is the concentration of betulin in the solution (mg/mL). The value of the correlation coefficient was 0.997.

The content of betulin in the initial samples was determined by chromatographic methods and amounted to 72.5%.

It is shown that after the first recrystallization, the obtained samples contain 84.6–95.4% betulin. After the second recrystallization, the betulin content is 88.2–98.3% (Table 4).

It was found that the optimal solvents for recrystallization are *n*-nonane (recrystallization temperature 150 °C) and *n*-decane (recrystallization temperature 174 °C). When using lower boiling solvents, such as *n*-octane (recrystallization temperature 125 °C), the yield of betulin decreases, while a significant amount of it remains in the fraction of insoluble impurities. This is due to the fact that the solubility of betulin in hydrocarbon solvents largely depends on temperature and decreases with its decrease. Thus, at temperatures of the cleaning process below 150 °C, it is necessary to increase the amount of solvent for the complete dissolution of betulin, while losses in the mother liquor increase. During recrystallization from higher boiling solvents, such as undecane (recrystallization temperature 195 °C), the quality of the resulting betulin decreases due to its partial tarring. It has been established that the optimal use is the use of solvents that allow the recrystallization process to be carried out in the temperature range of 150–180 °C; in this case there is practically no thermal degradation: betulin quantitatively passes into the liquid phase and, when cooled, it precipitates quantitatively. At the same time, the loss of betulin in the uterus is 1.6–1.8%.

It was found that the use of hydrocarbon solvents containing aromatic components, such as kerosene and nephras, leads to a decrease in the yield of betulin (74.0–75.1%) and to a decrease in its purity (85.4–84.6%). This is due to the fact that the solubility of betulin in aromatic solvents is higher than in aliphatic solvents, and the fraction of impurities insoluble in aliphatic hydrocarbons partially dissolves in aromatic solvents, polluting the resulting product.

Thus, in [65] the recrystallization of the total ethanol extract of betulin from high-boiling aliphatic hydrocarbons was investigated. It was shown that this method is convenient for the purification of betulin from accompanying impurities, and allows to obtain betulin containing up to 98% of the basic substance, suitable for further use in organic synthesis without additional purification. The implementation of this method showed that the loss of betulin during recrystallization is 1.6–1.8%.

It was established that the best result is achieved when using high-boiling aliphatic hydrocarbon solvents, such as *n*-nonan and *n*-decane. It was found that the presence of aromatic components in the solvent reduces the purity of the resulting product and its yield. 

Previously, an effective method of betulin extraction was proposed—thin-film vapor-phase extraction [61,62], which allows to intensify extraction by carrying out the process in a thin layer when the birch-bark layer is irrigated with extractant condensate, as well as due to an increased concentration gradient of extractive substances in the extractant — birch bark system. In addition, the same authors showed that a convenient way to purify betulin is the recrystallization of its extracts from high-boiling aliphatic hydrocarbons [65,66].

In a continuation of these studies, the goal of one of the works [67] was to develop a technology that combines the production of betulin in the form of a total ethanol extract and its subsequent purification by recrystallization, allowing to obtain a product with a purity of 88.2–98.3%, suitable for further use in organic synthesis. Figure 5 shows the technological scheme of betulin production developed by the authors.

The feedstock was pre-dried to a constant mass at a temperature of 105 °C to remove water. The residual moisture content of birch bark was 1.8%. Then, the dried birch bark was crushed in a knife mill to particles of 3–5 mm in size.

The technology of the extraction and purification of betulin was worked out on a laboratory installation. The extractor design was proposed earlier [61,62] and is shown in Figure 6.

The extractor is an extraction column made of quartz glass with an internal diameter of 90 mm and a height of 800 mm (1), on top of which a return refrigerator (2) is attached, and on the bottom, there is a 5–liter receiver cube (3) equipped with a nozzle for entering sharp water vapor. The receiver cube is heated by a heater (4). The extraction process is carried out as follows: as the extractant vapors are heated, they pass through a layer of crushed birch bark, condense in the reverse refrigerator (2) and return to the extraction column through the valve (5). The condensate irrigates the crushed birch bark, passing through its layer, is enriched with extractive substances and flows into the receiver cube. Then, the extractant evaporates again and returns to the extraction column, and the dissolved extract precipitates in the form of a granulate. The course of the extraction process is controlled by the presence of a dry residue in the draining condensate (6) at the outlet of the extraction column.

After drying and crushing, the prepared birch bark (1) and rectified alcohol (2) are loaded into the E-1 extractor, where the extraction process is carried out. Here, 86% ethanol was used as an extractant, which is due to the peculiarities of rectification of an aqueous alcohol solution. In addition, the presence of a small amount of water in alcohol leads to an increase in the extraction coefficient [64]. The duration of extraction was 4 h.

After the extraction is completed, the queen cell (4) is sent for planting in a container E-1, which is carried out by diluting the total extract with water. To remove alcohol from birch bark, acute water vapor (3) is fed into the receiver cube and the water–alcohol fraction (6) is distilled from the birch bark in an extraction column for subsequent regeneration. Regenerated alcohol is reused. The spent birch bark (5) is discharged from the column and can be used for the further extraction of fatty acids from it.

After planting, the suspension is directed to the F-1 filter, the sediment on the filter is washed with water, and the liquid phase (8) is directed to alcohol regeneration. The solid residue (7) is dried at a temperature of 105 °C in the dryer C-1.

Then, the dry extract (10) is recrystallized from the decane. We have previously shown that the recrystallization of betulin-containing extracts is convenient to carry out in an environment of high-boiling hydrocarbon solvents, such as nonane or decane, or mixtures thereof [65,66]. In addition, to reduce the cost of the process, it is promising to use commercially available isoparaffins of the C10-C12 fraction with a distillation temperature of 173–193 °C, for example, nefras I2–190/320. According to the technological scheme, the dry total extract (10) is loaded into a heated container E-2, dissolved in a boiling hydrocarbon solvent (11), and the resulting solution is subjected to hot filtration on an F-2 filter to separate insoluble impurities (12). The hot filtrate containing betulin is placed in a Kr-1 crystallizer and slowly cooled with cooled water to a temperature of 5–10 °C. The precipitated betulin is separated by filtration on the F-3 filter and washed with a light-boiling hydrocarbon solvent, for example, heptane. It is economically feasible to use aliphatic petroleum solvents of the composition C5-C7, for example, nefras P1–65/75 or nefras C3–94/99. The manifold (14) is directed to the regeneration of hydrocarbons. The solid residue after washing (13) is dried in a vacuum dryer C-2 at a temperature of 100–105 °C and the finished product, (15)—betulin, is obtained. Hydrocarbon vapors (16) are captured and sent for regeneration.

The material balance obtained for one technological cycle is shown in Table 5, Table 6, Table 7 and Table 8.

To analyze the quality of the obtained betulin, the HPLC method was used. The compounds were identified by NMR spectroscopy (see the Appendix A). The analyses were carried out according to the methods described earlier [65,66].

In the proposed technology of thin-film vapor-phase extraction, the approximation to the equilibrium state for the intensification of the extraction process is achieved by a constant supply of fresh solvent to the interface of the birch bark/extractant phases. In addition, the achievement of equilibrium is accelerated by the counterflow of the extractant along the height of the column.

Another important difference of the developed technology is that the process of birch-bark extraction and its subsequent stripping (solvent removal) is carried out in one device, which significantly increases the fire safety of production.

The proposed method, in contrast to classical extraction methods, for example, percolation, allows to achieve high degrees of extraction of the target component with a relatively low specific alcohol loading per cycle, i.e., the ratio of extractant to bark is minimal. This circumstance significantly improves the technical and economic indicators of the production process.

It is convenient to use aliphatic high-boiling hydrocarbons for the recrystallization of betulin-containing extracts. It was previously shown [63] that the use of solvents such as lower aliphatic alcohols, chloroform, acetone, ethyl acetate, and dichloromethane for recrystallization leads to the formation of sufficiently stable solvate complexes with betulin, which makes it difficult to isolate it as an individual compound.

During recrystallization from aliphatic high-boiling hydrocarbons, it is possible to avoid the formation of solvate complexes of betulin. In addition, the solubility of betulin in hydrocarbon solvents largely depends on temperature, and the use of solvents, with the help of which the recrystallization process is carried out in the temperature range of 150–180 °C, allows you to quantitatively transfer betulin into the liquid phase, and when cooled, it is quantitatively planted, which significantly reduces the loss of betulin in the mother liquor (1.6–1.8%). At the same time, the first recrystallization makes it possible to obtain a product with a purity of 84.6–95.4%, after the second recrystallization, the content of betulin 88.2–98.3%. According to the proposed technology, the yield of purified betulin is 91.4 g per 1000 g of the original birch bark.

Thus, the proposed technology for producing betulin by vapor-phase thin-film extraction makes it possible to intensify the extraction process by conducting the process in a thin layer at the interface of phases when the birch bark layer is irrigated with extractant condensate, as well as due to an increased gradient of concentrations of extractive substances in the extractant–birch bark system. As a result of subsequent purification by recrystallization from high-boiling aliphatic hydrocarbons, the resulting samples contain 84.6–95.4% betulin. After the second recrystallization, the betulin content is 88.2–98.3%. At the same time, the loss of betulin in the manifold is 1.6–1.8%.

According to the literature data [68,69,70,71], the use of a mixture of solvents can lead to an increase in the yield of extractive substances from birch bark. It has been shown that prolonged extraction with methanol provides the best yield of triterpenes [72].

Even taking into account patents, data on the effectiveness of the extraction and purification process of betulin and its derivatives is insufficient from the standpoint of the purity of the target product [23,42,43,44,59,72,73,74,75,76,77,78,79,80,81,82,83,84].

Thus, for the two-component ethanol-water extractant, it was found that optimal extraction conditions are achieved at a temperature of the liquid phase in the receiver of 83–85 °C and a temperature inside the extraction column of 82–84 °C; mass fraction of ethanol in the receiver is 22%. If these parameters are observed, the extract is obtained in the form of a granular precipitate, convenient for subsequent processing. With a decrease in the concentration of ethanol in the receiver, the extract is a flake-like mass, which complicates subsequent filtration. If the alcohol content is higher than 22%, a fine suspension is formed.

In the case of using an immiscible solvent with water, for example, hexane, the ratio of components does not significantly affect the composition and appearance of the product. In this case, coagulation occurs at the interface of hexane–water liquids during the regeneration of a light-boiling solvent.

Thus, the method of the thin-film vapor-phase extraction of plant raw materials has been developed on the example of birch bark. The main regularities of the influence of the nature of the light-boiling component, extractant on the composition, degree of extraction and form of the extracts obtained have been established. It is shown that the intensification of the extraction process is caused by the process in a thin layer when the birch-bark layer is irrigated with an extractant condensate, as well as due to the increased gradient of concentrations of extractive substances in the extractant–birch bark system.

The disadvantages of most of the known methods of obtaining betulin include the need for preliminary separation of bark into birch bark and bast, which is time-consuming and energy-consuming. In addition, only up to 20% of the bark mass is processed, since the outer layer of the bark accounts for 16–20%.

In order to increase the yield of extractive substances and reduce the duration of extraction, intensification methods are used. Some of them improve the hydrodynamic modes of the extraction process, others are based on the use of the mechanical, chemical and mechanochemical activation of raw materials. Sometimes, a positive effect is achieved by the fractional extraction of tree bark with organic solvents of different polarity.

Currently, all available methods for producing betulin are labor-intensive and require high costs to purify the resulting substance from impurities, so the main task remains to develop an effective method for obtaining pure substances for their use, primarily in medicine, and their modification with the formation of new compounds.

Various methods of the isolation and purification of betulin, mainly from birch bark, have been described. Simple extraction with organic solvents is the easiest way to obtain biologically active substances. Methanol, ethanol, propane-2-ol, *n*-heptane or *n*-hexane, ethyl acetate and its mixtures with ethanol and water, dichloromethane, a mixture of chloroform/dichloromethane/methanol, a mixture of ethanol and aqueous alkali, butane-1-ol, toluene, petroleum, limonene and others can be used as a solvent [28,30,59,72,73,74,75,76,77]. An example of the use of imidazole-based ionic liquids as effective extractants is known [78]. The extraction process, as a rule, is significantly accelerated when the birch bark is crushed or kibbled. There are methods for activating extraction using physical principles: ultrasonic treatment, activation by steam or superheated steam, or microwave irradiation [23,29,79,80]. It is claimed that supercritical (fluid) extraction with carbon dioxide, a mixture of methanol, ethanol or acetone, or extraction after esterification of betulin to its esters is a more complex method than simple extraction [23,80,81]. The method of betulin sublimation [82] at atmospheric pressure or using high vacuum and temperature has been given.

Recrystallization and/or various column chromatography methods are the main methods of the purification of betulin from extracts [73,75,76,77,83,84].

A relatively simple and cheap method of separating betulin from birch bark and purifying it to a very high purity (>99%) was proposed in [23]. The procedure for obtaining pure betulin after extraction consisted of the following sequential techniques:Removal of acids and other impurities with Ca(OH)_2_;Removal of lupeol by benzene extraction;Recrystallization from ethanol;Removal of residual (partially colored) impurities on silica gel in chloroform;Recrystallization from ethanol solution.

Despite the use of benzene and chloroform, the authors believe that the proposed method of the purification of betulin is environmentally friendly, since environmentally hazardous solvents are almost 100% recycled by distillation and can be reused in further purification processes or used for other purposes. It is believed that the resulting solid by-products of the bark are a potential source of suberin, whereas the fraction after acid removal is a rich source of betulinic acid, and the fractions after benzene treatment are a valuable source of lupeol.

Aqueous solutions of Na_2_CO_3_ and NaOH are widely used in extraction processes to create highly alkaline conditions and increase the yield of phenolic compounds [85,86].

In the development of previous studies, the solubility and purity of extractive substances from the outer bark of birch in petroleum ether at its boiling point were investigated [87]; subsequent works [88,89] showed that the use of fractional crystallization during extraction in a specially designed apparatus for intensive mass transfer can be a promising, inexpensive and simple method of simultaneous concentration of lupeol in the mother liquor and purification of betulin from mixtures of triterpenes (Figure 1).

The pretreatment of Na_2_CO_3_ is in line with the concept of environmental protection, since a polar solvent such as ethanol is non-toxic with respect to some targeted applications of betulin and its derivatives, where extracts obtained in petrochemicals are not allowed, for example, in the production of cosmetics, food additives or pharmaceuticals. It was found that the degree of extraction significantly decreases with the increasing age of the trunk. In addition, a higher betulin content was obtained from the outer bark collected from 15-year-old trunks of each birch species. The content of lupeol in extracts of the outer bark of silver birch was almost three times lower than in extracts of fluffy birch. Pretreatment of the outer bark of birch with hot water and aqueous solutions of Na_2_CO_3_ improved the quality of the extract obtained using polar solvents. A certain fraction of monosaccharides and phenolic compounds was isolated; as a result, the content of combined betulin and lupeol in ethanol extractives increased from 67.7% for untreated outer birch bark to 99.0% after 3 h of treatment with an aqueous solution of Na_2_CO_3_. There are two ways to obtain extracts rich in triterpene, without the content of phenolic compounds and monosaccharides: (1) loading the above substances from the outer bark of birch by pretreatment with an aqueous solution of Na_2_CO_3_, followed by extraction of target particles with polar solvents in which triterpene is highly soluble; or (2) the use of non-polar solvents in which these substances do not dissolve. The pretreatment of Na_2_CO_3_ conforms to the ecological concept by using polar solvents for further extraction. They are less toxic or non-toxic in relation to the production of cosmetics, food additives or pharmaceuticals.

It was previously shown that, in most cases, synthetic derivatives demonstrated a higher level of biological activity than the starting betulin [90]. Therefore, the production of high-purity betulin as a basis for synthesis is of great interest for scientific and industrial purposes, although the complete elimination of impurities is a complex technological task. The use of conventional green polar solvents (for example, alcohols) does not provide the selective extraction of betulin. The resulting extracts usually contain a mixture of triterpenoids and significant amounts of polar and low-polar impurities. The use of green solvents of intermediate polarity leads to an increase in the content of triterpenoids in extracts and a decrease in energy consumption in the extraction cycle, but does not lead to the production of pure betulin [91]. The content of betulin in extracts usually does not exceed 55–75%, depending on the raw materials and solvents used. Traditional methods of isolation and purification of betulin from birch bark [23,44,73] are mostly multi-stage, material- and/or labor-intensive, and involve the use of toxic reagents and solvents.

A method of purification of betulin from polar and nonpolar impurities using benzene and chloroform [23] and a combined method including alkaline purification and separation of impurities from betulin using aromatic hydrocarbon (xylene) [90] have been proposed.

It was found that the use of the “green” limonene solvent provides selective extraction of betulin from birch bark under conditions of microwave activation [29]. The resulting effect was achieved due to the better solubility of triterpenoids at a high temperature in lemon, good crystallization of betulin during cooling and insignificant extraction of polar impurities. According to the authors, an additional advantage is that hydrogenated monoterpenes are resistant to oxidation processes.

When developing methods for extracting betulin from birch bark, they are mainly guided by two principles:To ensure the selectivity of extraction, while first of all excluding the extraction of polar satellites (sugars and phenols);The use of inexpensive and environmentally friendly reagents and solvents in the process of obtaining pure betulin.

In the development of the above work [29], a sufficiently effective and environmentally friendly method for obtaining pure betulin using “green” monoterpenes was developed [91]. In this paper, approaches to obtaining pure betulin by treating birch bark with extracts using green low-polar solvents (pinene, turpentine), their hydrogenation products and limonene are considered. It was found that the use of monoterpenes for processing birch bark during boiling with a reverse refrigerator and under conditions of microwave activation makes it possible to extract triterpenoids with insignificant leaching of polar impurities. Energy-saving extraction of betulin from birch bark with aqueous ethyl acetate, saponification of the extract and purification of betulin in limonene or hydrogenated monoterpenes made it possible to obtain betulin with a purity of 95–97% from the extract with high total yields (75–82%) and with effective solvent recycling (81–87%). The efficiency of the separation of high-polar impurities is ensured by their preliminary transformation into salts that are completely insoluble in monoterpenes, while low-polar impurities are removed by the crystallization of betulin in monoterpenes. The use of hydrogenated monoterpenes, pinane and hydrogenated turpentine in the processing of extracts eliminates the accumulation of impurities and allows the efficient recycling of used solvents. The use of green monoterpene solvents in the process of the isolation of triterpenoids is considered as a promising and energy-saving alternative to traditional methods using toxic solvents and labor-intensive chromatographic purification methods. Further research of green low-polar solvents from natural raw materials for the extraction and purification of individual plant metabolites seems promising, since such technologies can be competitive from the point of view of economic feasibility [92,93].

## 4. Conclusions

This article considers economically feasible methods of betulin isolation. One of these methods is the microwave activation method. It is shown, by the example of betulin isolation from Kyrgyz birch (*Betula kirghisorum*), that the use of microwave activation reduces the extraction time by 15–20 times compared to traditional isolation methods. 

Thin-film vapor-phase extraction method is considered, which allows to intensify extraction by carrying out the process in a thin layer when the birch-bark layer is irrigated with extractant condensate, as well as by increasing the concentration gradient of extractive substances in the extractant–birch bark system.

A relatively simple and cheap method is proposed for separating betulin from birch bark and purifying it to a very high purity (>99%) by removing acids and other impurities using sodium hydroxide, followed by the removal of lupeol by benzene extraction with further recrystallization from ethanol.

Obtaining high–purity betulin as a basis for the creation of medicines is of great interest for scientific and industrial purposes, although the complete elimination of impurities is a complex technological task, the solution of which seems promising, since such technologies can be competitive from the point of view of economic feasibility.

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
