# Peer review of "Methods of Betulin Extraction from Birch Bark"

_molecules, 2022, doi:10.3390/molecules27113621_

Round 1

Reviewer 1 Report

The review of the manuscript entitled “Methods of betulin extraction from birch bark”. This article concerns a purpose to betulin (lup-20(29)-ene-3β,28-diol), which  is  a  pentacyclic  triterpene  alcohol  that  can  be  obtained  from  birch  bark. As it is commonly known, betulin  has  a  wide  spectrum  of  biological  and  pharmacological  activities. Nevertheless,  it  exhibits  very  poor  water solubility that clearly reduces its bioavailability. Some topics undertaken are interesting, but the relevance of the study in general has not been explained. The question is what is the originality of this review study and their scientific significance. Moreover, mentioned investigations more specifically the evaluation of data available in databases, is not particularly original and several major improvements are still needed in order to achieve a relevant manuscript. After reading this manuscript, some comments and qualifications came to mind as follows.

  • Abstract, there is no significant information. A big mistake is the lack of sufficient data about aims of work, main assumptions of the work, achievements, new approach in the field of betulin extraction methods and conclusions. The content should change significantly.
  • It is unacceptable for the same sentences to appear in different parts of the manuscript (page 1, lines: 9, 25).
  • Likewise, in the introduction glaring is the lack of sufficient information and this part requires some revision. Reference [1] was not included.
  • This study contains many inaccuracies and omissions. There are no relevant references to tables and figures!? In details: page 1, line 379: Figure 5 shows a chromatogram of a betulin sample obtained… it should be mentioned Figure 6. The NMR spectrum (Figure 7) requires a detailed description and assignment to the structure of a tested compound. Figures are generally of poor quality and need to be improved.
  • The conclusions part has been omitted (why?). Any scientific work should contain important elements of novelty and a new approach to the discussed problems. This aspect is certainly missing from the reviewed work. What are the significant differences from other scientific works in this field? Moreover, please identify advantages and disadvantages of the proposed methodology, discuss the possibilities of the obtained results use and what are the outlooks for the future?
  • Please pay attention on the References, at the moment are not in accordance with the requirements of the journal.

Summary:

The subject of the manuscript falls within the scope of Molecules. Nevertheless, due to the lack of essential elements of scientific work, the reviewed manuscript is recommended to publish after major revision.

Author Response

Dear Reviewer,

Thank you for your valuable feedback. Your suggestions have been taken into account. The drawings have been changed. Inside the drawings, the font was changed to match the font of the article itself. A literature reference was added 1. The list of references was adjusted in accordance with the requirements of the journal. Relevant conclusions on isolation methods have been added. The article's  "conclusion" part was added.

The article presents the principles of birch bark extracting by known methods, their disadvantages are indicated. Then examples of extraction by more progressive methods are given, for example, microwave processing in an ultra-high-frequency field. Its positive aspects are indicated.  The extraction method - thin-film vapor-phase extraction is presented. Its advantages are indicated.

Reviewer 2 Report

Demets et al reviewed Betulin Extraction. This manuscript can contribute to explaining the general contents to researchers in the related field, but I think the current manuscript is not enough for publication.

  1. Although this manuscript is a review of the extraction of Betulin, it will be improved if you describe more about the characteristics and application of the substance for Betulin.

  2. Abstract and Introduction are very similar. In addition, the Introduction does not contain enough information about Betulin.

  3. Authors should add missing references throughout the manuscript.
    e.g. )line 25-38: ‘Betulin was discovered more than two centuries ago. However, despite the constant interest and active research activity, no active pharmaceutical preparations based on betulin and other triterpene components of birch bark have yet been developed and produced. It should be recognized that to date, the hopes associated with betulin and its derivatives for the introduction of medicines based on them into medical practice have not yet been crowned with success, with the exception of a number of drugs that are currently undergoing various phases of clinical trials. Despite the fact that many promising areas of application of betulin and its derivatives have been established, the progress of their ap- plication in human life has significantly slowed down due to the lack of industrial, economically feasible availability of them as a starting material. One of the reasons for this state of affairs is the lack of reliable and comprehensive engineering solutions for processing birch bark. Despite the reliable methods available in the literature for isolating betulin from birch bark, the development of more advanced technologies for its isolation, purification and identification remains the object of close attention of many researchers.’
    line 43-48: ‘Betulin 1 (betulin, betulinol, birch camphor, lupendiol) is a crystalline organic substance discovered in 1778 by Toviy (Johann Tobias) Lovitz, who by sublimation isolated a white compound from birch bark and for the first time described its chemical properties. in 1831, the famous chemist of that time D. Mason gave it the name betulin (from Latin betula), which is also contained in birch tar and is a white resinous substance that fills the cavities of cork tissue cells on birch trunks and gives it a white color”

  4. The font style inside the figure should be unified.

  5. Figures 1, 4, 5, 6, 7 and8: Was it the author's own drawing? The references are required for each figure legend. Some drawings are of very poor quality for publication. I suggest redrawing.

  6. Figure 3: If the author wants to summarize and emphasize this part, I suggest replacing it with a table.

  7. References do not match the journal format.

  8. Region and country information is missing from the author's affiliation.

  9. It is necessary to review the manuscript in English by a native speaker
    .

Author Response

Dear Reviewer 1,

I have taken into account your comments. The abstract was re-composed. The "Introduction" part of the article has been slightly changed. Information about betulin was added. Figures 1,4,5, 6, 7 are the author's own drawings. Since these drawings are from original sources, they are not completely readable when copied, so they were redrawn. Scheme 1 is original.

Figure 3 is a listing of known extraction methods.

The list of references has been adjusted according to the requirements of the journal.

Round 2

Reviewer 1 Report

Review (R2) of manuscript number Molecules-1709682

The revised manuscript in some parts has been improved according to previously comments. My questions have been answered, but not at all. Nevertheless, I think that the manuscript in its present form can be published.

Please pay attention: please verify some weaknesses in the text edition of a final version (pages: 1, 4, 8/9).

Reviewer 2 Report

The authors have addressed the reviewer's concerns. I support the publication of this revised manuscript.